# Are 2.5D approaches superior to 3D deep networks in whole brain segmentation?

**Saikat Roy**[*1]                                                    SAIKAT.ROY@DKFZ.DE
**David Kügler**[*2]                                                  DAVID.KUEGLER@DZNE.DE
**Martin Reuter**[2,3,4]                                              MARTIN.REUTER@DZNE.DE

[1] *Division of Medical Image Computing, German Cancer Research Center, Heidelberg, Germany*

[2] *Image Analysis, German Center for Neurodegenerative Diseases, Bonn, Germany*

[3] *A.A. Martinos Center for Biomedical Imaging, Massachusetts General Hospital, Boston, MA, U.S.*

[4] *Department of Radiology, Harvard Medical School, Boston, MA, U.S.*

**Editors:** Under Review for MIDL 2022

## Abstract

Segmentation of 3D volumes with a large number of labels, small convoluted structures, and lack of contrast between various structural boundaries is a difficult task. While recent methodological advances across many segmentation tasks are dominated by 3D architectures, currently the strongest performing method for whole brain segmentation is FastSurferCNN, a 2.5D approach. To shed light on the nuanced differences between 2.5D and various 3D approaches, we perform a thorough and fair comparison and suggest a spatially-ensembled 3D architecture. Interestingly, we observe training memory intensive 3D segmentation on full-view images does not outperform the 2.5D approach. A shift to training on patches even while evaluating on full-view solves these limitations of both memory and performance limitations at the same time. We demonstrate significant performance improvements over state-of-the-art 3D methods on both Dice Similarity Coefficient and especially average Hausdorff Distance measures across five datasets. Finally, our validation across variations of neurodegenerative disease states and scanner manufacturers, shows we outperform the previously leading 2.5D approach FastSurferCNN demonstrating robust segmentation performance in realistic settings. Our code is available online at github.com/Deep-MI/3d-neuro-seg.

**Keywords:** deep 3D convolutional nets, whole brain segmentation, deep ensemble.

## 1. Introduction

2.5D approaches to volumetric segmentation of neuroanatomy limit the input space visible to their predictive models. However, models like FastSurferCNN (Henschel et al., 2020) use stacked slices while ensembling on 3 orthogonal views to achieve remarkable accuracy on 95 structures in less than 1 minute. In contrast to segmentation tasks with limited labels such as lesion, brain tumor, or segmentation of less granular structures, the large number of classes is a distinguishing challenge for neuro-segmentation. In theory, 3D volumes with added spatial context provide advantages for deep neural networks (Mehta et al., 2017). Standard 3D deep learning models such as the 3D-UNet (Çiçek et al., 2016), VNet (Milletari et al., 2016) for volumetric segmentation require significant GPU memory for even

---

[*] Contributed equally

moderately large volumes when segmenting a large number of classes. For example, the training of a standard V-Net with a $256 \times 256 \times 256$ volume for segmenting 79 structures (as in this work) requires over a 100 GB of memory with model and loss gradients *sharded* across multiple GPUs. To overcome large memory requirements, models rely on 3D patches (Moeskops et al., 2016; Mehta et al., 2017; Li et al., 2017; Dolz et al., 2018; Wachinger et al., 2018) extracted from whole volumes, at the cost of losing relevant spatial information. In contrast, SLANT (Huo et al., 2019) introduces the idea of specializing individual (spatially localized, SL) networks, i.e. one network for a specific sub-region. AssemblyNet (Coupé et al., 2019) expanded on the idea increasing the number of different models to 250.

In addition to previously mentioned methods, numerous 3D segmentation approaches have been suggested in the last few years for segmenting large number of structures in whole brain segmentation (de Brebisson and Montana, 2015; Roy et al., 2017; Jog et al., 2019; Roy et al., 2019). However, superior performance across multiple datasets of the 2.5D FastSurferCNN leads us to investigate – *Are 2.5D approaches superior to 3D deep networks for whole brain segmentation*? FastSurferCNN is fast, accurate and data-efficient in its training. 3D approaches, on the other hand, face generalization difficulties due to very large parameter counts, show infeasible memory utilization for a large number of classes, and can exhibit limited performance outside experimental conditions.

In this work, we analyze and compare the performance of 2.5D and 3D segmentation methods. We observe 2.5D models outperform all state-of-the-art 3D models. We (re)establish 3D networks as the performance leader with the following key methodological improvements: 1) the training in a randomized patchification scheme, 2) Self-Ensembling, and 3) Spatial Ensembling. Training on random patches reduces the memory footprint and puts less constraints on consistent patch boundaries while at the same time allowing the transfer of model weights to full-view evaluation with very limited performance loss. In Self-Ensembling, we build an ensemble from one model by changing the patch location. Spatial Ensembles consist of multiple models specialized to specific brain regions similar to Spatially Localized networks originally proposed by SLANT (Huo et al., 2019). Specifically, the limited performance of SLANT, nnUnet and full-view VNet leads us to suggest an improved method for training Spatially Localized networks using our previously learnt global representations. We also evaluated multiple architectural modifications of the backbone VNet without finding strong differences. After these improvements to the training and evaluation scheme including self-ensembling and spatial ensembling strategies, we find the performance of 3D models again superior to 2.5D for whole brain segmentation with 95 different structures. Throughout all results, we maintain a uniform training dataset for fair comparison in a large whole brain segmentation benchmark. Finally, we validate our method's improvements across disease and MRI acquisition diversity in comparison to the 2.5D benchmark.

## 2. Methodology

### 2.1. Learning a patch-based Network

Directly training on full-view inputs for even relatively small input volumes is challenging for 3D deep convolutional networks such as the modified VNet (Milletari et al., 2016) used as a backbone network here. Addressing this, patch-based networks train a model on smaller

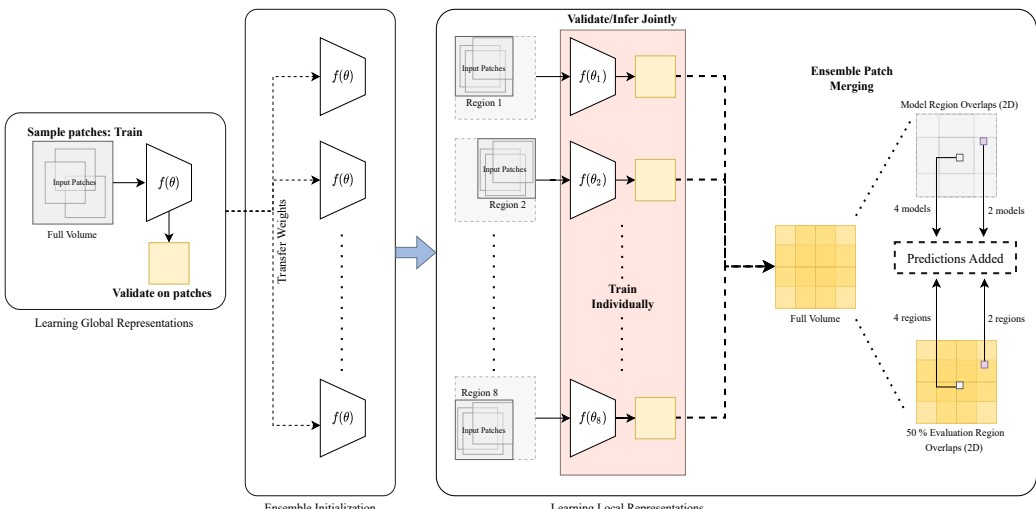

Figure 1: 2D illustration of our Spatially Localized deep network ensemble, where sub-networks are initialized with the global representations learnt on the full volumes. Sub-networks, with overlapping input regions larger than their field-of-view, are fine-tuned to learn local representations. Validation (as well as inference) is performed jointly by spatially ensembling outputs from sub-networks to emphasize full-volume predictive performance instead of those of individual networks.

sub-volumes. In this work, we sample input patches with a uniform distribution across patch positions and *without* additional padding at the volume boundary. However, this results in non-uniform voxel sampling skewed towards the image center during training. To formalize, let the top-left corner point of a random patch be $(x, y, z) \sim \mathcal{P}_{\text{uni}}^3$ from a uniform distribution $\mathcal{P}_{\text{uni}} = (L - l + 1)^{-1}$ between 0 and $L - l$ with the image and patch dimensions $L$ and $l$. The *voxel sampling density*, which is the probability that image voxel $(i, j, k) \in [0, L-1]^3$ is contained in a random patch, can be estimated by the patch indicator function $X_{ijk}(x, y, z)$, which is 1 inside the patch and 0 outside.

$$\mathbb{E}[X_{ijk}] = \sum_{x,y,z=0}^{L-l} \mathcal{P}_{\text{uni}} X_{ijk}(x, y, z) = \frac{P(i)P(j)P(k)}{(L - l + 1)^3} \text{ with } P(u) = \begin{cases} u + 1 & \text{if } u \in [0, l-1] \\ l & \text{if } u \in [l, L-l-1] \\ L - u & \text{if } u \in [L-l, L-1] \\ 0 & \text{otherwise} \end{cases} \quad (1)$$

In the training loop, the voxel sampling density may affect the network to focus on regions with high density, which is the brain positioned at the image center.

## 2.2. Learning an Ensemble of Spatially Localized Networks

To improve the predictive performance over single model 3D deep networks, we explore Spatially Localized (SL) networks. In contrast to one model applicable to all image regions (Global Representation), we obtain an ensemble of 8 SL-models (Local Representations) each associated with one image octant, i.e. network weights are allowed to specialize to the

different brain regions in their respective image octant[1]. To train the ensemble (and in contrast to SLANT), we initialize SL-models with the model learnt with patches from the whole image and fine-tune on patches from the image octant. In order to achieve overlapping regions of specialization, we sample patches by the patch location such that voxels at the octant boundaries are shown to multiple SL networks albeit with a different position in the respective patch (see Figure 1). We train the SL models collectively instead of in isolation: Per epoch, we iterate over the image octants and train each model independently on patches from its octant. To decide whether the updated octant-model should be committed to the ensemble permanently, we temporarily replace one corresponding octant model in the ensemble and determine the full-image validation performance of the ensemble (Spatial Ensembling with an Average Ensemble Size of 3.375, see Section 2.3).

Training the models sequentially minimizes GPU memory allocation. The pre-training allows the sub-networks to converge rapidly transitioning to fine-tuning. Instead of specializing SL-networks to a very explicit region of the volume (a limitation of SLANT), we specialize networks to fuzzy and overlapping regions. Consistent with fuzzy specialization, we remove the potentially erroneous and expensive registration to the MNI-template eliminating the exact and normalized position of the head in the volume in favor of more flexible networks. We provide the Algorithm 1 in the Appendix.

## 2.3. Ensembling and Evaluation Strategies

While full-volume (FV) inference provides one prediction per voxel, patch-wise evaluation enables two ensembling concepts to generate multiple predictions per voxel: 1. Self-Ensembling – *one model* predicts for different patches – and 2. Spatial Ensembling – *multiple networks* are specialized per image region and the appropriate network is chosen by the patch location. Finally, we aggregate predicted probabilities by class-wise sum and voxel-wise softmax. A simplified 2D illustration of Spatial Ensembling is shown in Figure 1.

## 2.4. Datasets

For comparability within methods and generalization across multiple scenarios, FastSurfer (Henschel et al., 2020) compiled a mixed-source dataset with dedicated splits for training, validation, and testing ($N = 140$, 20 and 1374, respectively) as detailed in Table 1. As in FastSurfer, the training set (validation set) exclusively consists of $256^3$ T1w images of 1 mm isotropic voxel size from ABIDE-II, ADNI, LA5c, OASIS1 and OASIS2 (only MIRIAD). In addition to in-dataset validation on ADNI, OASIS1 and MIRIAD (non-overlapping subject splits), we test whether the models generalize to unseen datasets (external validation) on HCP and THP. These datasets feature various age ranges, scanners, disease states and field strengths. Reference segmentations are obtained with FreeSurfer V6.0 and the "Desikan-Killiany-Tourville" atlas for parcellation (Fischl et al., 2002; Klein and Tourville, 2012). Following FastSurfer's approach, we merge the resulting 95 regions into 78 (foreground) labels for training and reconstruct the original 95 labels in postprocessing prior to evaluation.

---

1. We provide code at github and, for persistent availability, archive model weights of both the Global Representation (unensembled_model) and Local Representations (ensembled_model_<*index*>) online.

## 2.5. Architecture

We experimented with 8 architectural modifications to our backbone VNet including multiple architectural ideas from published work. However, we could not find significant differences between various top-performing architectures. We include an overview of results in the Appendix B. For the final (backbone) architecture, we replace each standard $5 \times 5 \times 5$ kernel with two $3 \times 3 \times 3$ kernels in a modified VNet (Szegedy et al., 2015). The batch size of 1 constrains memory usage. GroupNorm (Wu and He, 2018) increases stability in training with a small batch size. While we experimented with smaller patch sizes, only 3D input patches of size $128 \times 128 \times 128$ employed by each deep CNN in this work feature sufficient spatial context in training.

## 2.6. Baselines

FastSurferCNN is the representative 2.5D method while the full-view VNet trained on full volumes is the first 3D baseline. A 3D-UNet architecture, generated and trained with the nnUNet (Isensee et al., 2021) pipeline on a manual split, is selected to provide an architecturally improved 3D-UNet baseline. SLANT-27 (Huo et al., 2019) shares fundamental similarities with our proposed Spatial Ensemble of SL models. It represents a state-of-the-art, 3D ensemble-based baseline for neuroimage segmentation. Training and validation sets are fixed for all baselines and the proposed method.

## 3. Results and Discussion

We compare the performance our method and baselines in three experiments: A cross-dataset evaluation, an in-depth analysis of the effect of popular spatially localized ensembles and a generalization analysis of disease states and scanner manufacturers.

### 3.1. Segmentation accuracy

First and without considering the method presented here, we analyze whether 2.5D approaches are superior to 3D deep networks in whole brain segmentation. To this end, we compare the 2.5D FastSurferCNN with three popular 3D baselines: VNet, SLANT-27 and nnUNet (3D-UNet). In a cross-dataset evaluation, we find FastSurferCNN consistently outperforms all baselines 3D approaches with respect to both Dice Similarity Coefficient (DSC) and Average Hausdorff Distance (HD) across five datasets, despite its limited spatial context. Specifically, FastSurferCNN achieves higher accuracy and more consistent results under same-training dataset conditions (see Section 2.4).

However, when improving 3D networks by our 3D methodology, our method achieves average DSC of $0.878 \pm 0.032$ and an Average HD of $0.172 \, \text{mm} \pm 0.091 \, \text{mm}$ outperforming all 3D baselines[2] by at least 0.03 and 0.076 mm (44% error reduction), respectively (see Figure 2). Our approach outperforms the 2.5D FastSurferCNN also achieving high consistency across all datasets. Consequently, 2.5D methods are not superior to 3D deep networks. The significantly worse results of SLANT may have three causes: 1. architectural differences (e.g.

---

2. The backbone V-Net implementation achieves best results and is therefore selected as the reference, margins to other methods are larger.

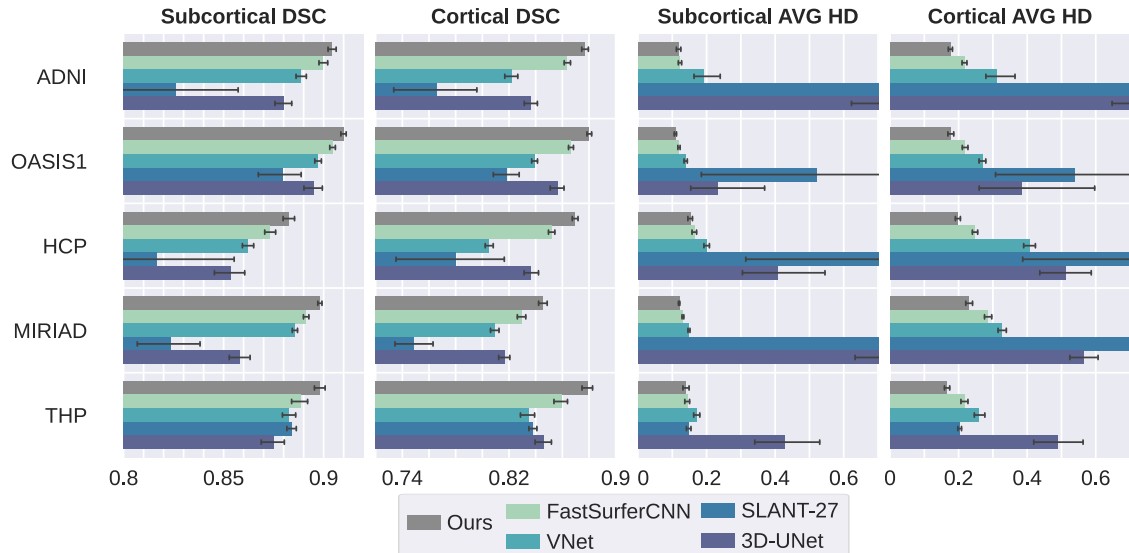

Figure 2: Dice similarity coefficient (DSC) and Average Hausdorff distance (HD) of cortical and subcortical structures in each dataset. The proposed network outperforms all baselines including FastSurferCNN significantly for all evaluations (Wilcoxon Signed-Rank test with Bonferoni correction, $p < 10^{-14}$). In fact, our method outperforms all baseline methods on 98% and 97% of subjects in the evaluation set (DSC and Average HD, respectively).

ours has more layers), and 2. unreliable registration to MNI space, and 3. lossy interpolation during both training and inference (Henschel et al., 2022).

### 3.2. The impact of Ensembles

Since several recent whole brain segmentation publications (Huo et al., 2019; Coupé et al., 2019; Henschel et al., 2020) employ ensembles at its methodological core, we analyze how ensembling impacts the segmentation performance. Our experiments include 1. a full-volume evaluation ("Ours (FV)"), 2. a Self-Ensembled model ("Ours (w/o SL)"), and 3. a Spatial Ensemble of eight SL-models ("Ours"). "Ours (FV)" and "Ours (w/o SL)" are based on the same training and weights – the global representation used to initialize the SL-models. In our analysis, we exclude ensembles generated from various data splits. We plot distributions of both DSC and Average HD for our method featuring different "degrees of ensembling" together with the four baselines in Figure 3. Since we want to analyze in how far ensembling contributes to the performance, we introduce the "Average Ensemble Size". For it, we count the number of predictions per image voxel (the per-voxel Ensemble Size) and average across the image, since it varies across the image. For patch-based evaluation schemes, this primarily depends on the overlap and image and patch dimensions (see Table 3). Methods with an Ensemble Size of 1 do not use ensembling.

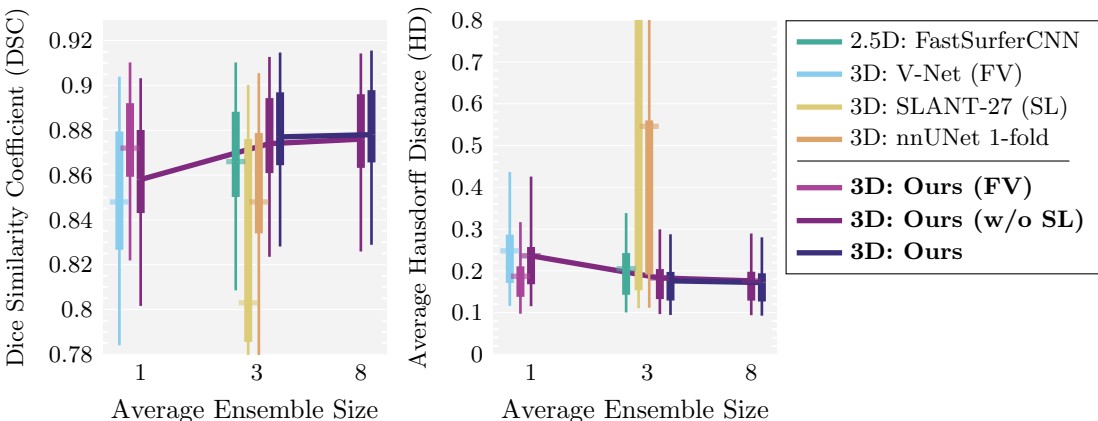

Figure 3: Evaluation of multiple Ensemble sizes (five datasets-average): Our method shows limited dependence on the Average Ensemble Size while outperforming the state-of-the art in Dice Similariry Coefficient and Average Hausdorff Distance. Note, at Ensemble Size 1 patch-based evaluation loses local context at the patch border reducing performance. Running the same model in full-view mode (Ours FV) recovers most of the performance. At 1.24 mm, the Average Hausdorff Distance of SLANT-27 is outside the view. FV: Full-View, SL: Spatially Localized.

We find the performance of our method increases with increasing ensemble size (see both "Ours (w/o SL)" and "Ours" in Figure 3). However, the dependence of the performance on the ensemble size is very limited for both "Ours (w/o SL)" (Self-Ensembling) and "Ours" (Spatial Ensembling). While "Ours (w/o SL)" has a significantly reduced performance at Ensemble Size 1, this phenomenon should not be misinterpreted as a trend, since predictions on the boundary of the patch do not gain the spatial context across the patch boundary. Disabling patch-based evaluation for this model, i.e. inference on the full volume in full-view-mode (FV), recovers most of the performance loss (Dice difference 0.002 between "Ours (FV)" at Ensemble Size 1 and "Ours (w/o SL)" at Ensemble Size 3.375) – even without retraining on the full-view dataset. Training on patches and evaluating on full-view volumes might prove a feasible strategy to mitigate the memory requirements of high class number and large volume segmentation applications.

Spatial Ensembling improves segmentation results by up to 0.003 Dice as indicated by the comparison of "Ours (w/o SL)" and "Ours". This relatively small difference diminishes the value of Spatial Ensembling in practice. Furthermore, Spatial Ensembling does not support the transfer of patch-based models to full-view during evaluation.

### 3.3. Generalizability across diagnosis and scanner manufacturers

Meta-data available on the ADNI dataset enables exploratory analysis of the DSC of cortical and subcortical segmentation performance extensively across diagnosis (Cognitive Normal (CN), Mild Cognitive Impaired (MCI), Demented (AD)) and scanner manufacturers (Siemens, GE, Phillips) (Figure 4). There is a small decrease in performance on scanner

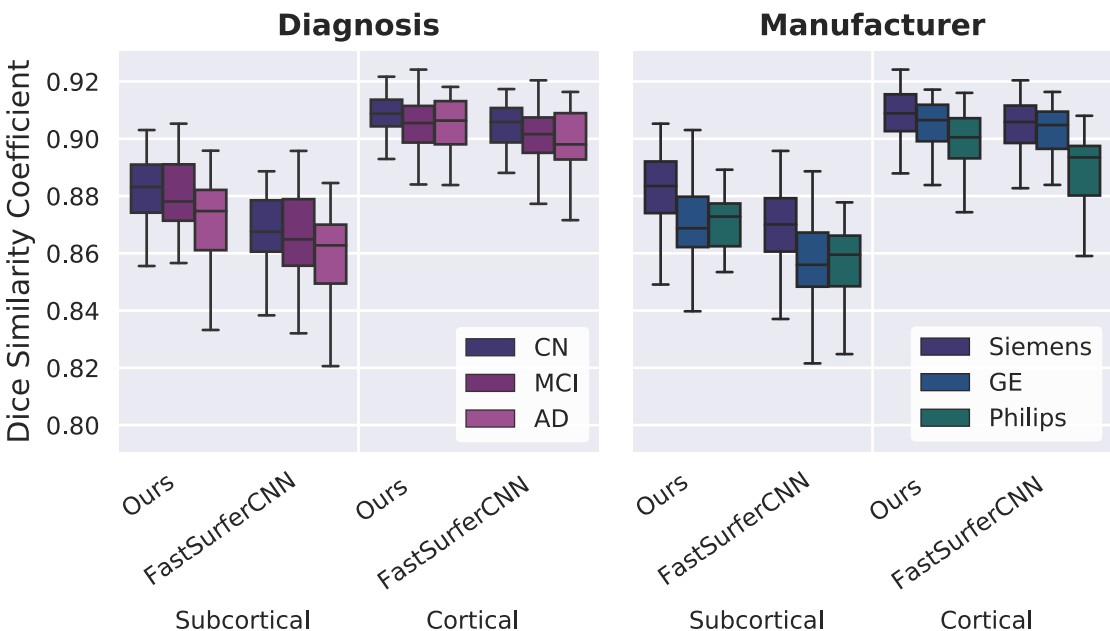

Figure 4: Dice Similarity scores of cortical and subcortical structures across diagnosis and device manufacturer on the ADNI dataset for FastSurferCNN and our method. We observe superior performance across individual groups.

manufacturers (GE, Phillips) less-represented in the training set, a trend similar to Fast-SurferCNN. Our method improves upon FastSurferCNN's performance in both categories for cortical and subcortical structures. This makes the proposed technique suitable for multi-study, multi-device, multi-site datasets. The improved accuracy across disease states ensures reliable performance in studies involving neurodegenerative disorders. While small reductions in DSC are observed with disease progression similar to FastSurferCNN, our method demonstrates better mean DSC in all disease states.

## 4. Conclusion

In this work, we answer whether 2.5D networks are superior to 3D deep networks. While prior to this publication, this seemed to be the case, we identify several modifications of the training and evaluation to improve the performance of 3D models. This includes exploration of a 2-step training procedure with learning global representations followed by spatially-localized models, as well as, strategies for inference incorporating self-ensembling and spatial ensembling. Our 3D deep networks for whole brain segmentation with 95 structures outperform all baseline models in five evaluated datasets. Moreover, the training strategy solves GPU memory limitations while still allowing high-quality full-view evaluation relevant for integrated pipelines operating on full-view feature maps. The method also benefits from very fast inference times dominated by overhead operations, e.g., 1 second network inference plus 10 seconds constant overhead for full-view evaluation (see Table 4).

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

| Usage | Dataset | Scanner | 1.5T/3T | State | Age | Subjects |
|-------|---------|---------|---------|-------|-----|----------|
| **Training** | ABIDE-II (Di Martino et al., 2017) | Phillips | 3T | Autism/Normal | 20-39 | 20 |
| | ADNI (Mueller et al., 2005) | Phillips, GE, Siemens | 1.5T/3T | AD/MCI/Normal | 56-90 | 40 |
| | LA5c (Poldrack et al., 2016) | Siemens | 3T | Neuropsych/Normal | 23-44 | 20 |
| | OASIS1 (Marcus et al., 2007) | Siemens | 1.5T | Normal | 18-60 | 40 |
| | OASIS2 (Marcus et al., 2010) | Siemens | 1.5T | AD/Normal | 66-90 | 20 |
| **Validation** | MIRIAD (Malone et al., 2013) | GE | 1.5T | AD/Normal | 60-77 | 20 |
| **Evaluation (Accuracy)** | ADNI | Phillips, GE, Siemens | 1.5T/3T | AD/MCI/Normal | 58-85 | 180 |
| | HCP (Van Essen et al., 2012) | Siemens | 3T | Normal | 22-35 | 45 |
| | OASIS1 | Siemens | 1.5T | AD/Normal | 18-96 | 370 |
| | MIRIAD | GE | 1.5T | AD/Normal | 55-80 | 49 |
| | THP (Magnotta et al., 2012) | Phillips, Siemens | 3T | Normal | - | 5 |

Table 1: The dataset used in the training, validation and evaluation is a replica of the corresponding sections of the FastSurfer (Henschel et al., 2020) dataset. The number of samples in the Evaluation set is 1374 as some datasets contains multiple scans of subjects.

David C Van Essen, Kamil Ugurbil, Edward Auerbach, Deanna Barch, Timothy EJ Behrens, Richard Bucholz, Acer Chang, Liyong Chen, Maurizio Corbetta, Sandra W Curtiss, et al. The human connectome project: a data acquisition perspective. *Neuroimage*, 62(4):2222–2231, 2012.

Christian Wachinger, Martin Reuter, and Tassilo Klein. Deepnat: Deep convolutional neural network for segmenting neuroanatomy. *NeuroImage*, 170:434–445, 2018.

Yuxin Wu and Kaiming He. Group normalization. In *Proceedings of the European conference on computer vision (ECCV)*, pages 3–19, 2018. URL https://doi.org/10.1007/s11263-019-01198-w.

## Appendix A. Ensemble Training Algorithm

The ensemble training in this work uses the VNet as the base model. However, the algorithm is model-independent and may be adopted for any compatible deep neural network model. This is detailed in Algorithm 1.

## Appendix B. Architectural Modifications

Various modifications to the backbone VNet architecture are tried as presented in Table 2. The modifications all wrap around the main Encoder-Decoder macro-architectural style of the VNet without a complete overhaul. The focus is intended to be techniques that have resulted in improvements in past work from literature. The results here are evaluated as mean and standard deviations of dice similarity coefficients on the 78 non-background classes, without reconverting back into the 95 classes. It should be noted that the VNet with $3^3$ kernels is the backbone network used throughout the work. The general trend observed is that the architectural modifications do not provide any benefits to the segmentation performance.

---

**Algorithm 1:** *Volumetric Ensembling* using *tiled fine-tuning* and *ensembled validation*

---

**Input:** $L$: Loss function, $Tr$: Training set, $Vld$: Validation set $l$: Patch Size, $f$: neural network with weights $w$, pretrained as in Section 2.1 ;

**Initialization:**

$P_{main}$: List of $m$ nets with weights $w$, for each octant of 3D input space (default $m$: 8) ;

$P_{curr}$ : List of $m$ neural networks with weights $w$ as replica of $F_{main}$ ;

$F_{rng}$ : List of $m$ *effective receptive field* ranges (25% greater than the octant size) ;

$dice(f)$ : Dice score of $f$ evaluated on $Vld$;

**Algorithm:**

$DSC_{best} = 0$ ;

**while** *epochs not done* **do**

    $m_{cnt} = 1$ ;

    **while** $m_{cnt} <= m$ **do**

        **while** $(X, y) \in tS$ **do**

            Select a point $p_{ijk} \in F_{rng}[m_{cnt}] - l$ ;

            $\tilde{X} = X[p_i : p_i + l, p_j : p_j + l, p_k : p_k + l]$ ;

            $\tilde{y} = y[p_i : p_i + l, p_j : p_j + l, p_k : p_k + l]$ ;

            $E = L(\tilde{y}, F_{curr}[m_{cnt}](\tilde{X}))$         // Calculate loss on octant

            $\forall w \in F_{curr}[m_{cnt}]$, update $w = w - \eta \nabla_w E$      // update current pool

        **end**

        Insert $P_{curr}[m_{cnt}]$ *temporarily* into $P_{main}[m_{cnt}]$ ;

        $DSC_{curr} = \text{dice}(p_{main})$ ;

        **if** $DSC_{curr} \leq DSC_{best}$ **then**

            Revert changes to $F_{main}[m_{cnt}]$ ;

        **else if** $DSC_{curr} > DSC_{best}$ **then**

            Keep changes to $p_{main}[m_{cnt}]$ ;

            $DSC_{best} = DSC_{curr}$ ;

        $m_{cnt} := m_{cnt} + 1$;

    **end**

**end**

**Return:** $p_{main}$ as the trained *Spatially Localized (SL)* models

---

## Appendix C. Training

We train the full-view model for for 100 epochs, starting with a learning rate of $10^{-3}$ (and halving at 40, 80, 90 epochs) with the Adam optimizer (Kingma and Ba, 2014). To obtain an ensemble of Spatially Localized (SL) models, we branch out the training into eight different networks associating each with the vicinity of a corner (overlapping subvolumes of $192^3$ voxels). Each epoch, we train each SL model in serial and jointly validate the entire ensemble patches of 50% overlap. Here, we restart the learning rate at $10^{-3}$ (20 epochs), and reduce to $5/2.5/1.25 \cdot 10^{-4}$ (for 10 epochs each). Throughout, we use a combined loss function of median frequency-balanced cross-entropy and dice loss (Roy et al., 2017).

| Model Description | Params | DSC Mean ± S.D. |
|---|---|---|
| *Reference architectures* | | |
| FastSurferCNN | $1,799,223$ | $0.8696 \pm 0.0589$ |
| VNet with GroupNorm (full-view training) | $71,086,301$ | $0.8505 \pm 0.0617$ |
| *Architecture modifications (patch-based training)* | | |
| Backbone VNet with GroupNorm and $3^3$ Kernels instead of $5^3$ (Szegedy et al., 2015; Simonyan and Zisserman, 2014) | $16,319,197$ | **0.8803** $\pm 0.0544$ |
| Backbone VNet with Symmetric filter factorization with two $3^3$ kernels instead of $5^3$ (Szegedy et al., 2016) | $30,241,282$ | $0.8800 \pm 0.0549$ |
| Backbone VNet with Asymmetric filter factorization with two sets of $(3 \times 1 \times 1)$, $(1 \times 3 \times 1)$, $(1 \times 1 \times 3)$ filters instead of $5^3$ (Szegedy et al., 2016) | $10,677,910$ | $0.8748 \pm 0.0561$ |
| Backbone VNet with Dense residual block as in DenseNet, with additive residuals (Huang et al., 2017) | $71,091,794$ | $0.8505 \pm 0.0708$ |
| Backbone VNet with Attention Block as in Attention UNet (Oktay et al., 2018) | $71,153,273$ | $0.8783 \pm 0.0536$ |
| Backbone VNet with Squeeze and Excite block as in SqueezeNet (Hu et al., 2018) | $71,262,110$ | $0.8767 \pm 0.0548$ |
| Backbone VNet with Concurrent 3 axis 2.5D and 3D paths with 75% of feature maps in 3D path (de Brebisson and Montana, 2015) | $64,104,868$ | $0.8784 \pm 0.0543$ |
| Backbone VNet with only 75% of feature maps included in each block | $45,679,866$ | $0.8797 \pm 0.0542$ |

Table 2: Architectural modifications to the VNet (with Group Norm) architecture trained as in Section 2.1 and evaluated with 75% overlapping patches of size 128. Results are obtained as average Dice Similarity Coefficients on 78 classes without remapping.

## Appendix D. Baseline Methods

Baselines are DNN models whose results serve as existing state-of-the-art solutions in the area of neuroimage segmentation. Most of these models have wide acceptance in the research community as standard segmentation networks or are distinguished by recent publication to top venues with exceptional benchmarks. There are 5 models used by us as benchmarks and a brief description of each of them are provided in the following subsections.

### D.1. FastSurferCNN

FastSurferCNN (Henschel et al., 2020) was introduced as a 2.5D technique for neuroimage segmentation. Three 2D encoder-decoder based deep CNN architectures were trained individually on one of three orthogonal views (coronal, sagittal and axial) of the brain. The architecture of all 3 models were kept the same with the input to each model being a *thick slice* of 7 consecutive slices from the corresponding view. The architecture used Maxout-based Dense blocks (as opposed to concatenation-based blocks) to achieve a model with less paramters. The final segmentation output was obtained after view aggregation of the segmentation outputs provided by the individual networks. FastSurferCNN provides fast, robust and accurate segmentations and is used as the non-3D benchmark for our evaluations.

### D.2. SLANT

3D patch-based segmentation techniques usually train and predict on sub-volumes of the input space followed by post-hoc fusion of the partial predictions to produce a complete segmentation. SLANT (Huo et al., 2019) was centred on the idea of decoupling spatial context during feature search. This was done by registering each volume to the MNI305 (Evans et al., 1993) template during postprocessing, prior to learning or inference. This was subsequently followed by training sub-nets for sub-regions of the neuroimage — one variant was trained on non-overlapping patches and another on overlapping ones. In theory, a successful registration exposes each subnet to a single consistent brain/neuroimage region, thereby reducing or even eliminating the need to *learn* the positional context. The networks were retrained on our dataset and the original code was kept as much as possible. All networks were trained individually with non-coupled losses (in terms of a global prediction) and inferences from each network was fused using majority label voting. This was followed by inverse registration to the original data space to provide the final segmentation. The SLANT-27 variant (overlapping regions) provided the strongest performance in the original publication and has been adopted as a 3D benchmark for our evaluation.

### D.3. 3D-UNet: nnUNet single-fold

Neural network based solutions to segmentation problems usually vary architecturally from each other, with the variations ranging from minor to significant. The nnUNet (Isensee et al., 2021, 2018) framework was designed to adapt the the block architecture of a standard robust architecture, specifically, the UNet (Ronneberger et al., 2015; Çiçek et al., 2016) and customize an encoder-decoder model based on those blocks on a provided dataset. This allowed for some standardization in block design while still customizing architecture to a particular problem and associated data. The 3D-UNet architecture recommended by the nnUNet framework on our training data was used as one of the baseline models for the purposes of our evaluation.

### D.4. VNet

The VNet (Milletari et al., 2016) was introduced as a full (or near full-volume) 3D deep learning-based segmentation architecture for medical imaging. It is designed as an encoder-decoder type model with residual blocks on individual layers as well as residual connections

| Model | AES (OL, PS) | DSC ± S.D. (↑) | AVG HD ± S.D. (↓) |
|---|---|---|---|
| 2.5D: FastSurferCNN | 3 | 0.866 ± 0.036 | 0.205 ± 0.098 |
| 3D: VNet (FV) | 1 | 0.848 ± 0.041 | 0.248 ± 0.160 |
| 3D: SLANT-27 | 3.375 (50%, 128) | 0.803 ± 0.178 | 1.110 ± 5.061 |
| 3D: nnUNet 1-fold | 3.375 (50%, 128) | 0.848 ± 0.057 | 0.546 ± 1.110 |
| 3D: Ours (FV) | 1 (0%, 256) | 0.872 ± 0.033 | 0.187 ± 0.010 |
| 3D: Ours (w/o SL) | 1 (0%, 128) | 0.858 ± 0.036 | 0.236 ± 0.169 |
| 3D: Ours (w/o SL) | 3.375 (50%, 128) | 0.874 ± 0.032 | 0.183 ± 0.127 |
| 3D: Ours (w/o SL) | 8 (75%, 128) | 0.876 ± 0.032 | 0.176 ± 0.098 |
| 3D: Ours | 3.375 (50%, 128) | 0.877 ± 0.032 | 0.176 ± 0.098 |
| 3D: Ours | 8 (75%, 128) | 0.878 ± 0.032 | 0.172 ± 0.091 |

Table 3: Numeric values for Figure 3 Evaluations, AES: Average Ensemble Size, OL: Patch Overlap, PS: Patch Size

| Model | Ens. Size | Training Time (GPU time) | Epochs | Inf. Time (secs) |
|---|---|---|---|---|
| 2.5D: FastSurferCNN | 3 | 5 hours | 30 | 60 |
| 3D: VNet (FV) | 1 | (4 days 20 hours)×4 (requires 4 GPUs) | 100 | 15 |
| 3D: SLANT-27 | 3.375 | (1 day 2 hours) × 27 | 100 | 900 |
| 3D: nnUNet 1-fold | 3.375 | 3 days 12 hours | 1000 | N/A |
| 3D: Ours (FV) | 1 | | | 11 |
| 3D: Ours (w/o SL) | 1 | 2 days 7 hours | 800 | 11 |
| 3D: Ours (w/o SL) | 3.375 | (same model) | | 25 |
| 3D: Ours (w/o SL) | 8 | | | 42 |
| 3D: Ours | 3.375 | 1 day 19 hours (+ 3D: Ours (w/o SL)) | 50 | 40 |
| 3D: Ours | 8 | (same models) | | 133 |

Table 4: The total training and per sample inference times of each model under experiment. Our 3D:Ours models also include the training time of the non-SL model from which they were initialized

between corresponding layers from the encoder to the decoder. A slightly modified variant of the VNet, incorporating Group Normalization (Wu and He, 2018), is used as a 3D benchmark for our experiments. This particular evaluation is noteworthy as the modified VNet is also the backbone architecture for most of the techniques introduced and explored in this thesis.

## Appendix E. Training and Inference Time

Table 4 indicates the total training time and inference time per sample for all models. The training time is indicated in GPU-hours on 32GB Tesla V100s along with the number of epochs each model was trained. The training dataset is the same for all models. The inference times are represented per sample in seconds. The discrepancy in inference times between 3D:VNet(FV) vs 3D:Ours(FV) is due to the former needing heavy model sharding

to allow training on full input volumes. This also increases the inference time slightly, though it can be optimized to be similar to the other model.

## Appendix F. Statistical Tests on Model Performance

The statistical significance tests between our method with the best performance and 4 baselines from Figure 2 are performed pairwise for *DSC* and *Average HD* metrics with groupings for cortical and subcortical structures in addition to combined results on whole brain structures. The Wilcoxin Signed-Rank test is used at 5% significance with Bonferoni correction for multiple comparisons. For every comparison performed, we obtain the corrected $p < 10e - 14$ indicating statistical significant differences between the performance of our method in comparison to competing baselines across all groupings and metrics.

