# OpenReview forum: "Are 2.5D approaches superior to 3D deep networks in whole brain segmentation?"
_MIDL.io/2022/Conference — MIDL 2022_

### Official Review · Reviewer_dvQ4 · 2022-01-21

**Confidence:** 4
**Preliminary Rating:** 4
**Recommendation:** Oral, Poster

**Summary:**

The paper presents a methodological development of modifying 3D CNN networks for brain structure segmentation tasks. These modifications lead to better performance of 3D CNN than 2.5D CNN when evaluated on several publicly available datasets. Without these modifications, 2.5D CNN (FastSurfer) performed better than other popular 3D CNNs (nnU-Net, 3D V-Net, etc.). Overall the paper presents a good methodological development and a validation study for the brain structure segmentation task. Some concerns regarding the implementation details are listed in the weakness and detailed comment section.

**Strengths:**

* The paper is clearly written; and with good motivation.
* Methodological development is well motivated and clearly written,
* Fig.1 is really helpful in understanding spatially localized network
* Experiments are well defined, and results are presented well.
* Evaluation on publicly available datasets with different diseases type and different scanner manufacturers shows the generalizability of the proposed method.
* Results provided in the appendix helps in showing that a lot of effort was made to make the baseline as good as possible.

**Weaknesses:**

* The authors argue that one of the key contributions is low GPU memory requirement due to patch-based training compared to full volume-based training. Although, this comes at an increased inference cost when a similar patch-based technique is used inference. The authors should mention this when stating the advantage of patch-based training. It would be helpful if the authors mention total training and inference times for all models in Table-3 (appendix).
* In Sec.:2.2, the authors mention that models are trained collectively instead of in isolation by only committing a model that improves the ensemble performance (tested at the end of epoch by validation metrics). I am not sure how this is achieved? As per my knowledge, there are eight different models, all of which are trained sequentially. A little bit better explanation for this would be helpful.
* Similarly, in Sec.:2.2, it is mentioned that GPU memory costs are minimized by training the models in sequence. This also comes at the increased training time. It would be helpful if the authors mentioned this in the paper.
* It is not clear why 95 regions were consolidated into 78 regions during training? (Sec.:2.4)
* The paper is missing comparison against a V-Net architecture trained on patches used for learning global representation. The first stage of the proposed method. This comparison would help evaluate the usefulness of the learning local representation using spatially localized networks (stage-2 of the proposed method).
* It is unclear why the ensemble size is 3 for 3D SLANT-27 and 3D nn-Unet in figure-3.
* In Fig:4, a breakdown of the model performance based on scanner manufacturer is provided, but there is no discussion regarding the same in the paper's main text. It is unclear why the performance decreases for GE and Phillips scanners compared to Siemens scanners as MRIs from all three scanner manufacturers are used in training (Table-1).
* It would be helpful if the authors provided statistical significance analysis to compare different models.

**Deanonymize Review:**

no

**Detailed Comments:**

* References for all datasets are missing from the main text of the paper.
* Authors use all datasets with ground-truth obtained using FreeSurfer software. Maybe validating the proposed method using MICCAI 2012 multi-atlas labeling challenge dataset [1] would be a good idea, as they provide manually corrected labels for 30 subject MRI. Similarly, the MindBoggle dataset [2] is also a good option as they provide 101 MRIs with manually corrected labels.
* Figures 2,3, and 4 could improve, if authors provide minor ticks on X-Axis, in addition, to already provided major ticks. In the current state, it is hard to interpret the exact values of each bar as they fall between provided two values. This is necessary as the authors mention exact values for some methods and also differences between the two methods in the text. It is not easy to verify this from the figures.
* Authors divide the results in Fig:2 by site and scanner manufacturers, but no discussion regarding the same is provided in the paper. Maybe the authors would like to remove this part of the figure. The saved space could help in making other things more clear.
* Authors are missing some reference for brain structure segmentation papers which either tackles the problem using patch-based 3D CNN [3,4] or slice-based 2D CNN with enough spatial context [5].

1.  http://www.neuromorphometrics.com/2012_MICCAI_Challenge_Data.html
2. https://mindboggle.info/data.html
3.  Li, W., Wang, G., Fidon, L., Ourselin, S., Cardoso, M.J. and Vercauteren, T., 2017, June. On the compactness, efficiency, and representation of 3D convolutional networks: brain parcellation as a pretext task. In International conference on information processing in medical imaging (pp. 348-360). Springer, Cham.
4.  Dolz J, Desrosiers C, Ayed IB. 3D fully convolutional networks for subcortical segmentation in MRI: A large-scale study. NeuroImage. 2018 Apr 15;170:456-70.
5. Mehta R, Sivaswamy J. M-net: A convolutional neural network for deep brain structure segmentation. In2017 IEEE 14th International Symposium on Biomedical Imaging (ISBI 2017) 2017 Apr 18 (pp. 437-440). IEEE.

**Final Rating After The Rebuttal:**

5: Strong Accept

**Justification Of The Final Rating:**

The authors have sufficiently addressed all my primary concerns. There are still some suggestions that I have provided in the response, which could be helpful in the future more extended version of the paper. I have changed my recommendation to strong acceptance.

**Paper Type:**

both

**Questions To Address In The Rebuttal:**

Overall paper is well written with a good contribution. But it would be helpful if the authors answered all the questions raised in the weakness section. This would help in improving the clarity of the paper.

**Special Issue:**

no

---

### Official Review · Reviewer_RvGN · 2022-01-24

**Confidence:** 5
**Preliminary Rating:** 5
**Recommendation:** Oral

**Summary:**

The authors perform a thorough comparison and analysis of 2.5D and 3D methods for whole-brain region segmentation. While most literature suggests that 2.5D methos (such as FastserverCNN) are superior in performance to 3D methods, the authors propose some methodological improvement exploiting ensembling, resulting in a successful 3D method with improved performance. The method is tested on a large set of datasets, with attention for splitting datasets correctly and analyzing the effect of factors such as scanner vendor and diagnosis.

**Strengths:**

- The paper is very well organized and well written. The motivation is clear, and the work is novel and thorough
- Novelties in methodologies that improve 3D methods are: randomized scheme for extracting patches, self-ensembling and spatial ensembling
- Combines local patches with learning a global model.
- Prior work is adequately assessed, used very representative methods as baseline. Convincing analysis and key question in the field of neuroimage analysis.


**Weaknesses:**

I very much like this paper and do not have major weaknesses to mention. I like that this paper is placing a novel method in the context of a bigger question (2.5D vs 3D) and uses a thorough evaluation to both show the performance of the proposed methods as well as answer the question. Only weakness that I can think of is that improvement may be only incremental in a clinical context.

**Deanonymize Review:**

yes

**Paper Type:**

both

**Questions To Address In The Rebuttal:**

- The training set seem relatively small. Why is this chosen, and would increasing training set size change performance and conclusions?
- How do training times and memory compare between FastSurfer, patch-based and the proposed method?


**Special Issue:**

yes

---

### Official Review · Reviewer_t5Zk · 2022-01-24

**Confidence:** 4
**Preliminary Rating:** 5
**Recommendation:** Oral

**Summary:**

The authors set out to empirically analyse why whole brain segmentation approaches based on neural networks with full 3D inputs from the literature tend to produce worse results than Fast-surfer which is a 2.5D approach.

The authors propose three key adaptions to a 3D patch-based learning approach and show that these lead to improved performance of 3D full brain segmentation that performs better than the 2.5D approach. In particular this work introduces:
- randomized patch sampling
- self-ensembling
- spatial ensembling using 8 different models in different subregions of the volume with spatial overlap

Both self- and spatial-ensembling aggregate the votes from different models or regions at voxels that belong to different models or regions.

This adapted 3D approach is systematically compared a V-net, nnUNet, a spatially localized SLANT-27 model and the Fast-Surfer method on openly available MRI datasets, where a FreeSurfer 6.0 segmentation is used as the reference.

**Strengths:**

- I think this work addresses (and answers) a very important question for the neuroimaging community, as results of deep neural network-based segmentation have not been as impressive as in other fields and the success of 2.5D approaches is indeed somewhat counterintuitive. I would encourage the authors to extend this work (e.g. in a journal article) and to further work out the details and the exact mechanisms leading to better or worse performance of 3D vs 2.5D approaches in neuroimage segmentation.

- The results of the work are convincing and interesting.

- The authors provide a detailed and helpful appendix.

- Overall I think this is a solid and important contribution, that would be of considerable interest in the neuroimaging community.




**Weaknesses:**

The paper is written quite well in most parts, but some parts (e.g. the introduction, or the second part of 3.1.) lack clarity in my opinion. In particular, the part explaining the experiments varying the different ensembling types on p6-7were a bit confusing and should be revised.



**Deanonymize Review:**

no

**Detailed Comments:**

- For me a critical point would be code and model availability: To be of use to the neuroimaging community (at which I think this paper is addressed), it would be necessary to publish the code and pre-trained model(s) from this paper.

- The part explaining the experiments varying the different ensembling types were a bit confusing and should be rewritten.
	- How did you set up the experiments and what did you compare?
	- Maybe the "average ensemble size" isn't the best dimension to discuss the models. I found it a bit confusing somehow compared to just stating the category of each approach (e.g. with or without spatial ensembling)
	- It would be good to clearly explain what n=1 means for self-assembling.

- p7: "Spatial ensembling also improves segmentation results by up to 0.004 Dice as indicated by the comparison of “Ours (w/o SL)” and “Ours”. In consequence, we do not observe the performance improvements by spatial ensembling presented in previous publications."
     - I do not get the conclusion here, how does that conclusion from the preceding sentence?

- The discussion of the results is a bit too short.
	- e.g. what explains the huge deviations in performance of the 3D-UNet and SLANT-27 in Figure2?

- p4: "To this end, a set of 8 VNets is used with overlapping input sub-spaces. In contrast to SLANT, we train models collectively instead of in isolation by 1) only committing a model that improves the ensemble performance (tested at end of epoch by validation metrics) and 2) by sharing the initial global representation."
	- Is this collective training necessary to improve the performance of the method? Did you test this? Please explain the collective training. (It becomes clearer from the pseudocode, but the text is a bit unclear I think.)

- p4: "Reference segmentations are obtained with FreeSurfer V6.0 and the “Desikan-Killiany-Tourville” atlas for parcellation (Fischl et al., 2002; Klein and Tourville, 2012)."
	- What if FreeSurfer results are worse than predictions of your model? Did you observe this case? This might particularly be a problem for subcortical regions. Can you comment on this?

- p4: "We consolidate the resulting 95 regions into 78 (fore- ground) labels for training, reconstructing them in postprocessing prior to evaluation."
	- Why are the regions consolidated here and how?

- p8: "we identify several modifications of the training and evaluation to improve the performance of 3D models."
	- Please summarize the most important modifications here briefly.

- p8: " The method also benefits from very fast inference times dominated by IO operations."
	- Would be good to briefly explain this claim.

minor stylistic points:
- Style of writing can be improved in general, I give some results below
	- The introduction starts with a couple of sentences that do not really explain the claims or are a bit vague.
		- p1 "The task of segmenting a large number of classes is markedly different from those having few classes such as lesion, brain tumor, or segmentation of less granular structures." Is it harder or easier?
		- p2 "We observe 2.5D models outperform all state-of-the-art 3D models and improve 3D network performance significantly." Do 2.5D models improve 3D network performance or did the steps outlined in this paper on 3D models improve 3D network performance?
		- p4 "To test whether the models generalize to unseen datasets (external validation) evaluations cover HCP and THP in addition to ADNI, OASIS1, MIRIAD (always non-overlapping) with various age ranges, scanners, disease states and field strengths." is missing a verb
		- p5: "SLANT-27 (Huo et al., 2019) is selected as the baseline for a state-of-the-art volumetric ensemble method for neuroimages, as the proposed ensemble shares fundamental characteristics with it." Shares fundamental characteristics with what?



**Final Rating After The Rebuttal:**

5: Strong Accept

**Justification Of The Final Rating:**

The authors have responded to my comments and also revised the manuscript accordingly. Since the paper is a strong contribution in my view in particular for the Deep Learning in Neuroimaging community.

**Paper Type:**

both

**Questions To Address In The Rebuttal:**

The authors should discuss if they will make the code and models available to the community. This is a critical point in my opinion.
I also think that the style of writing should be improved and in particular part 3.1 should be revised.

**Special Issue:**

yes

---

### Meta-Review · Area_Chair_niqZ · 2022-02-18

**Recommendation:** Accept (Oral)
**Confidence:** 5

**Metareview:**

The paper presents a novel idea of a 2.5D segmentation approach considering the characteristics of medical imaging data. The practical importance and novelty of the proposed approach are well received by all the reviewers. Clear description and illustration are included appropriately in the paper. Some experimental details-related questions were raised by the reviewers, and they were well addressed in the rebuttal.

---

### Decision · Program_Chairs · 2022-02-28

Accept